# Clinical Activities, Contaminations of Surgeons and Cooperation with Health Authorities in 14 Orthopedic Departments in North Italy during the Most Acute Phase of Covid-19 Pandemic

**DOI:** 10.3390/ijerph18105340

**Published:** 2021-05-17

**Authors:** Alessandro Aprato, Nicola Guindani, Alessandro Massè, Claudio C. Castelli, Alessandra Cipolla, Delia Antognazza, Francesco Benazzo, Federico Bove, Alessandro Casiraghi, Fabio Catani, Dante Dallari, Rocco D’Apolito, Massimo Franceschini, Alberto Momoli, Flavio Ravasi, Fabrizio Rivera, Luigi Zagra, Giovanni Zatti, Fabio D’Angelo

**Affiliations:** 1Azienda Ospedaliera CTO-CRF Maria Adelaide, Università degli Studi di Torino, 10126 Turin, Italy; ale_aprato@hotmail.com (A.A.); alessandro.masse@unito.it (A.M.); alessandra.cipolla94@gmail.com (A.C.); 2Regional Health Care and Social Agency Papa Giovanni XXIII, 25127 Bergamo, Italy; cccastelli@libero.it; 3Department of Biotechnology and Life Sciences (DBSV), Università degli Studi dell’Insubria, 21100 Varese, Italy; delia.antognazza@gmail.com (D.A.); Fabio.DAngelo@uninsubria.it (F.D.); 4Fondazione Poliambulanza Istituto Ospedaliero, 25124 Brescia, Italy; francesco.benazzo@unipv.it; 5Azienda Ospedaliera Niguarda Ca’ Granda, 20162 Milano, Italy; bovefederico@yahoo.it; 6Ortopedia e Traumatologia, Spedali Civili Di Brescia, 25123 Brescia, Italy; axcasi@yahoo.com; 7Orthopaedics and Traumatology, Modena University Hospital, Università degli Studi di Modena e Reggio Emilia, 41121 Modena, Italy; fabio.catani@unimore.it; 8IRCCS Istituto Ortopedico Rizzoli, 40136 Bologna, Italy; dante.dallari@ior.it; 9IRCCS Istituto Ortopedico Galeazzi, 20161 Milano, Italy; roccodapolito@hotmail.it (R.D.); luigi.zagra@fastwebnet.it (L.Z.); 10Orthopedic Institute Gaetano Pini, 20122 Milan, Italy; massimo.franceschini@asst-pini-cto.it; 11Orthopaedics and Traumatology, San Bortolo Hospital, 36100 Vicenza, Italy; alberto.momoli@gmail.com; 12ASST-Melegnano-Martesana, Ortopedia di Cernusco sul Naviglio, 20070 Vizzolo Predabissi, Italy; flavioravasi@alice.it; 13Civil Hospital SS. Annunziata, 12038 Savigliano, Italy; rivgio@libero.it; 14Orthopaedics and Traumatology, University of Milano–Bicocca, 20900 Monza, Italy; giovanni.zatti@unimib.it

**Keywords:** orthopedics and traumatology, contamination, COVID-19, management, organization

## Abstract

*Background*: From 10 March up until 3 May 2020 in Northern Italy, the SARS-CoV-2 spread was not contained; disaster triage was adopted. The aim of the present study is to assess the impact of the COVID-19-pandemic on the Orthopedic and Trauma departments, focusing on: hospital reorganization (flexibility, workload, prevalence of COVID-19/SARS-CoV-2, standards of care); effects on staff; subjective orthopedic perception of the pandemic. *Material and Methods*: Data regarding 1390 patients and 323 surgeons were retrieved from a retrospective multicentric database, involving 14 major hospitals. The subjective directors’ viewpoints regarding the economic consequences, communication with the government, hospital administration and other departments were collected. *Results*: Surgical procedures dropped by 73%, compared to 2019, elective surgery was interrupted. Forty percent of patients were screened for SARS-CoV-2: 7% with positive results. Seven percent of the patients received medical therapy for COVID-19, and only 48% of these treated patients had positive swab tests. Eleven percent of surgeons developed COVID-19 and 6% were contaminated. Fourteen percent of the staff were redirected daily to COVID units. Communication with the Government was perceived as adequate, whilst communication with medical Authorities was considered barely sufficient. *Conclusions*: Activity reduction was mandatory; the screening of carriers did not seem to be reliable and urgent activities were performed with a shortage of workers and a slower workflow. A trauma network and dedicated in-hospital paths for COVID-19-patients were created. This experience provided evidence for coordinated responses in order to avoid the propagation of errors.

## 1. Introduction

Coronavirus disease 2019 (COVID-19) is an infectious disease caused by severe acute respiratory syndrome coronavirus 2 (SARS-CoV-2), a coronavirus first identified in December 2019 in Wuhan (China), which then rapidly spread all over the world, so much so that on 11 March 2020 the World Health Organization (WHO) declared it a pandemic. Globally, at the end of September 2020, there have been more than 70 million confirmed cases of COVID-19, including about 1.6 million deaths, reported to WHO [1]. In Italy, the first death due to COVID-19 was reported on 21 February 2020, and in mid-March 2020, the number of deaths rapidly increased, resulting in almost 50% of excess deaths from any causes in March 2020 [2]. On 10 March 2020, the Italian Government declared the national lockdown due to the high number of infected people and COVID-19 related deaths. The lockdown marked the beginning of the so-called “phase 1”, which lasted from 10 March 2020 to 3 May 2020. During that period, SARS-CoV-2 was spreading widely and not contained, moreover the resources of the national health system (particularly intensive care units, ICUs) were not enough and a triage for disasters was adopted [3,4]. As already described in detail by European societies for Orthopedics and Traumatology [5,6] as well as in Singapore [7], only patients requiring urgent or early orthopedic care (such as trauma or musculoskeletal tumors) were admitted to the hospital [8].

This new scenario has forced health systems to a complete reorganization of medical activities. Healthcare resources and medical staff were redirected to manage the infected patients; nonurgent activities were suspended or postponed, releasing resources and avoiding unnecessary exposure for patients and healthcare providers. Personal Protective Equipment (PPE) was a topic of great concern during the first wave. Firstly, there was scarce experience with COVID-19, secondly the availability of PPE forced a rationalization and often downgrading of the PPE, leading to uncertainty among the health care workers (HCWs) during a critical situation [9].

The aim of the present study is to assess the impact of COVID-19 pandemic on Orthopedic and Trauma care in Northern Italy, focusing on:(a)Hospital reorganization, in terms of flexibility (description of the organizational modifications, workflow and protocols changes), workload (volume and type) in the trauma and orthopedic units; prevalence of COVID-19/SARS-CoV-2 among patients; observance of the standards of care.(b)Effects on the medical staff: contamination and direct involvement in COVID-19 units, after the reorganization and during lock-down, eventually absence rate (due to contamination or reallocation).(c)Subjective perception of the pandemic from the orthopedic point of view, through an anonymous questionnaire among the directors of the involved centers.

## 2. Materials and Methods

The database was completed by 14 hospitals from major urban centers in Northern Italy, aimed to study the impact of SARS-CoV-2 on the orthopedic patient population (Table 1). The involved hospitals were: nine tertiary care hospitals and trauma centers (Bergamo, Papa Giovanni XXIII Hospital; Brescia, Spedali Civili; Milan, Niguarda Hospital; Modena, University Hospital; Monza, San Gerardo University Hospital; Pavia, IRCCS Policlinico San Matteo Hospital; Turin, Città della Salute e della Scienza; Varese, Circolo e Fondazione Macchi Hospital; Vicenza, San Bortolo Hospital); three orthopedic clinics (Bologna, IRCCS Rizzoli Institute; Milan, Gaetano Pini Institute; Milan IRCCS Galeazzi Orthopaedic Institute) and two secondary care hospitals (Vizzolo, Predabissi Hospital; Savigliano, SS Annuziata Hospital). Data are included only if they referred to the period of lock-down (the first “red” phase) during which the spread of SARS-CoV-2 was not controlled and only the in-patients were included.

### 2.1. Hospital Reorganization

Data regarding the workload of the 14 involved hospitals were extracted: number of patients, creation and type of dedicated COVID-19 operating rooms, procedures’ volume comparison between 2019 and 2020, patients’ demographic, fracture types, prevalence of COVID-19 and SARS-CoV-2, time from admission to surgery, timing and type of hospitalization and, particularly for proximal femur fractures classified as 31 according to AO-OTA classification, the survivorship at 1 month. The number of procedures was compared with the same period in 2019.

### 2.2. Effects on the Medical Staff

The same 14 Centers were asked to complete an anonymous survey, regarding the involvement of the orthopedic surgeons in COVID units and their contamination rates with SARS-CoV-2, plus other organizational details like the received on-field training, availability of PPE and workdays lost (Figure 1). The median leaves from work were calculated, considering sick leave and the isolation period without symptoms.

### 2.3. The Orthopedic Point of View

Each director of an Orthopedics and Traumatology Department was asked to fill a questionnaire, concerning his/her subjective viewpoint about communication and cooperation with the government, hospital administration and the other departments, along with an evaluation of the economic consequences for the Department. The questionnaire was intended only for the Heads of the involved Centers and was created by Haffer et al. [10] for a similar survey in Germany. It has been developed in cooperation with the Institute of Psychometry and Health Outcome Research of the Charité University Clinic (Berlin). After a literature review had been performed, the questions were developed by the first and senior authors in internal consensus meetings. The questionnaire was revised and approved by external coauthors and the Executive Board of the German Society of Orthopedics and Trauma Surgery. For the present study, the questionnaire was then neither translated from English nor validated in Italian [11], but adapted by 2 senior Authors (CC, ZL), in order to accommodate the differences between the Italian and German health systems [10].

The questionnaire (Figure 1) included individual questions and statements in five categories (government, medical authorities, hospital administration, hospital and department reorganization and perception of future scenarios) using a combination of a closed questions, open questions and 5-point Likert scale.

**Figure 1 ijerph-18-05340-f001:**
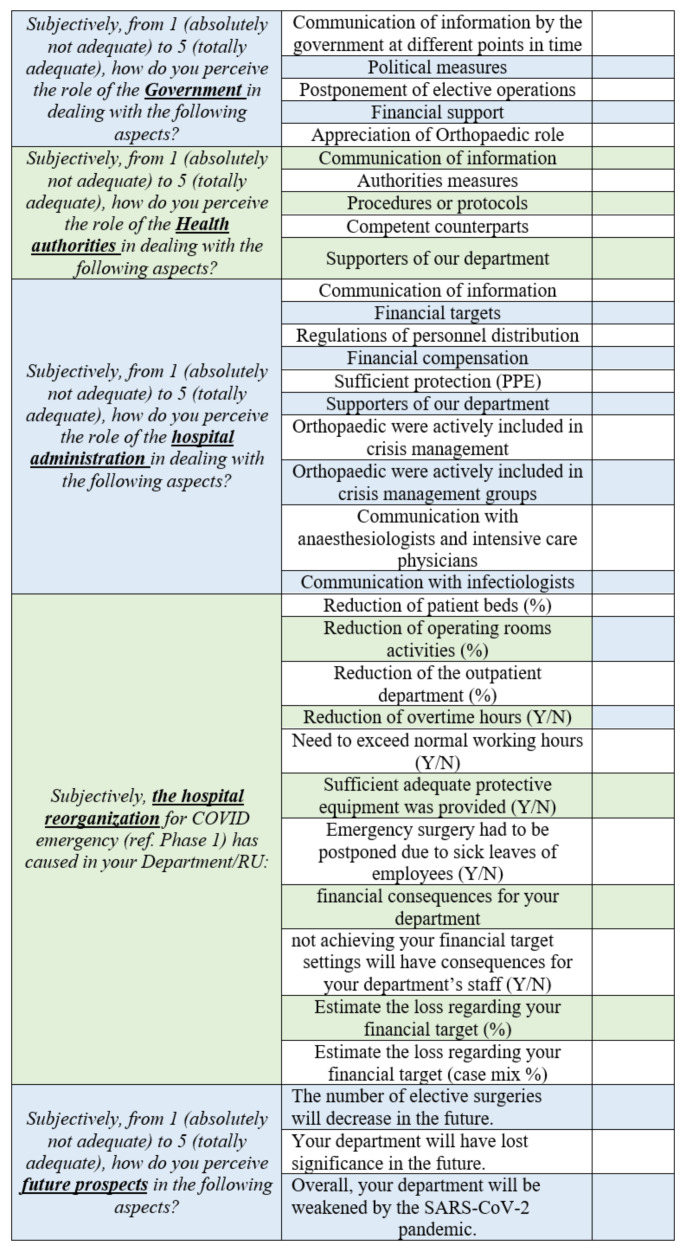
The questionnaire used in the study.

The study was conducted among the Directors of Orthopedics and Traumatology of the involved centers, concerning their subjective viewpoints about communication and cooperation with the Government, hospital administration and other departments, along with an evaluation of the economic consequences on the Department. In the first section (Government; 5 questions) and second section (Medical Authorities; 5 questions), interviewees were asked to rate on the Likert scale the role of authorities in emergency management and decisions made by the governors. In the third section (Hospital Administration; 10 questions) interviewees were asked about their hospital reorganization, the available resources, including the state of equipment and personal protective equipment (PPE) at their hospital. The fourth section (Department Reorganization; 11 questions) evaluated the impact of the COVID-19 pandemic on orthopedic and trauma surgery departments. The last section (Perception of future scenarios; 3 questions) allowed the responders to express themselves about future scenarios. This questionnaire was translated and adapted from a nationwide survey of the Orthopedic and Trauma Surgery in University Hospitals in Germany [10].

This study was conducted in accordance with the Institutional review board of the reference center (Ethical Committee, Bergamo; Ref. No. 31–21), which collected data from the participating centers in an aggregated and anonymous form. The study protocol was in accordance with the Declaration of Helsinki for human research.

### 2.4. Statistics

Sample distributions were tested for normality with the Kolmogorov−Smirnov test. Accordingly, normally distributed data were described with mean, standard deviation and CI95, whilst non-normally distributed and ordinal variables were described with median, interquartile range 25% and 75% (IQR_25–75_) or percentage, as appropriate. Statistical analyses were computed with Microsoft Excel (Microsoft Corporation, Redmond, WA, USA).

## 3. Results

### 3.1. Hospital Reorganization

All centers used dedicated COVID-19 operating rooms. Some 83% (10/12) of the hospitals organized dedicated COVID-19 wards and COVID-19 free ones, mostly with two patients/room. The 1390 patients were treated in 14 centers. Overall, the volume of surgical procedures was −73% (IQR_25–75_ −50 to −83) in comparison with the same period in 2019 (Table 1). The patients’ demographic data are summarized in Table 2. The mean time from admission to surgery overall was 57 ± 50 h (CI_95_ 52.5–61.5). Median hospitalization time was 7 days (IQR_25–75_ 3.25–11) days. On admission, 40% (554/1390) underwent a swab test for SARS-CoV-2 and 7% (37/554) resulted positive; 7% (91/1390) received medical therapy for COVID-19. Only 48% of patients who received medical therapy for COVID-19 (44/91) had a positive swab test and 3% (3/91) needed support in an Intensive Care Unit (ICU), 11% (10/91) in a sub-ICU. The frequency of the main fracture locations is reported in Table 3: 34% (477/1390) were proximal femur fractures in adults (AO-OTA 31), the latter was the most frequent fracture type (Table 2). In this group, the average time between admission and surgery was 2.3 ± 0.4 (CI_95_ 2.19–2.41) days in patients that were positive at COVID-19 tests and 2.8 ± 0.1 (CI_95_ 2.79–2.81) days in negative patients. The mortality at 30 days in AO-OTA 31 in adults was 23% (12/53; CI_95_ 0.13–0.36) in COVID-19 positive versus 5% (20/424; CI_95_ 0.03–0.07) in COVID-19 negative, with an odds ratio (OR) = 0.17 (CI_95_ 0.08–0.37); *p* = 0.008. The mean hospitalization was 14.7 ± 1.5 days for COVID-19 and 10.9 ± 0.3 days for non-COVID-19.

### 3.2. Effects on Medical Staff

In the present study, 323 orthopedic surgeons from 12 centers were included, 11% (37/323) developed COVID-19, without known long-term consequences. A further 14% (5/37) developed symptoms before the lock-down; among the other 32, 5/32 developed symptoms after working in COVID units (3/5 within < 6 weeks and 2/5 > 6 weeks after starting in COVID units). Further 6% (18/323) resulted in being asymptomatic carriers for SARS-CoV-2 on swab test, however only 3/12 centers performed screening on their employees (with a nasopharyngeal PCR swab test on SARS-CoV-2) during the observed period. In 11/12 hospitals, orthopedic surgeons were redirected in COVID units. On average, 14% of the staff were daily occupied in COVID units. Before the end of the survey, 45% of the whole staff of each center carried out at least one shift for one week. In 67% of the hospitals, shifts in COVID units and COVID-free areas were mixed (orthopedic surgeons had to accomplish duties in COVID-free areas while working in the COVID shift). Only in 1/12 centers was the activity planning intended to split the surgeons into COVID or COVID-free areas, but due to the urgent needs of the orthopedic department, one or more orthopedic surgeons were recalled to COVID-free duties while in COVID units. No one had instructions from the hospitals to avoid familiar contacts during the shifts in COVID units or to quarantine before coming back to “COVID-19-free” environments. The medical staff of 3/11 Departments of Orthopedics and Traumatology involved in COVID units were redirected to COVID units without specific training; all the others received a specific academic preparation, apart from one that received both practical and academic training. The personal protective equipment (PPE) for COVID-19 was insufficient in 4/12 hospitals, according to the personal evaluation of the Directors of Orthopedic/Trauma.

### 3.3. The Orthopedic Point of View

The main results of the survey among the heads of the Departments, are displayed in Figure 2, Figure 3, Figure 4, Figure 5 and Figure 6 and Table 4. The answers were mostly heterogeneous. With reference to the Government, the majority agreed or had a neutral position with the conduct, in opposition to medical Authorities: departments heads did not agree in terms of protocol clarity (not appropriate for 80%), appropriateness of measures and overall communication (50%). About hospital administration, the communication was appropriate for 58% and 80% refer to an adequate communication with other Departments (Anesthesiology and Infectious disease). Some 50% report a lack of involvement with the crisis unit and support to Orthopedics and Traumatology departments. There was a 73% reduction for operating rooms and 80% for outpatients’ activities: all but one center will not achieve their budget objectives in 2020.

**Table 4 ijerph-18-05340-t004:** Reduction of activities. Interquartile range (IQR), from 25th to 75th.

Variable	Median	IQR 25–75
Drop in number of beds	−55	35–76
Drop in activity in operating rooms	−73	50–83
Drop in outpatients	−80	65–90

## 4. Discussion

The SARS-CoV-2 pandemic is a global challenge, affecting the whole health system. In the present study, we refer to the first wave in Northern Italy, during the “red” phase and with severe and noncontrolled spreading of SARS-Cov-2. The need to urgently treat patients overwhelmed the heath system and hindered all other duties; neither detecting the carriers nor properly isolating the patients was possible.

To our best knowledge, this is one of the largest cohorts of treated patients and caregivers for orthopedics and traumatology during the COVID-19 pandemic reported till now. Different types of hospitals were involved, all active in orthopedics and traumatology during the lock-down. According to the type of survey, a part of the data is based on subjective evaluations.

Considering the present data gathered in Orthopedics and Traumatology, the following considerations can be made.

### 4.1. Hospital Reorganization

The volume of activity in orthopedics and traumatology was drastically reduced during the “red” phase: −73% operating rooms and −80% outpatients’ procedures. The reduction was intentional on one side (complete stop of elective surgery) and a consequence of the lock-down policy on the other side (e.g., less sport/vehicle/work accidents). Consequently, this reduction was more evident for orthopedics than traumatology. As observed by other Authors [14], during the first month of the pandemic, this reduction did not include the number of domestic accidents and fragility fractures [7,15,16]; our data confirm this observation [17]. Proximal femur fractures accounted for 34% of all inpatients and was the most common fracture type overall. In this group, the majority of patients had surgery within 48 h as indicated by international guidelines. Compared with 2019, the rate of patients operated on after 48 h went from +7% to −65% as stated by other Authors [17]. Even though in our work we did not extract these data referring to 2019, the treatment of proximal femur fractures in patients over 65 years is considered in Italy a work-flow and quality indicator for Health Structures and those data are collected by, analyzed and published by the National Healthcare Outcomes Program (PNE) of the Italian National agency for Regional Healthcare Service (AGENAS). According to the report of 2019, in Piemonte, Lombardia and Emilia Romagna, the patients treated within 2 days for proximal femur fractures ranged from 67% to 78%. Those data are referred to all proximal femur fractures treated in every hospital of the considered regions; thus, a high variability is expected concerning both the patients’ complexity and hospitals’ surgery volume [18,19]. Thus, according to our data, the time to surgery was similar to the nonpandemic period. Having a lower workload and a greater availability of operating rooms after the stop of elective surgery surely contributed to the flexibility and adjustment for new workflows, respecting the standard of care [20]. The necessity to keep a low workload is reflected in our data by the relative high time-to-surgery, even for non-COVID-19 patients. This increase was due to the time needed for SARS-CoV-2 screening when applicable, longer time needed to have an operating room, including dramatic decrease of operating room efficiency [20], less availability of anesthetists and nurses.

The first step was to reduce activity and perform only urgent duties, to set beds and staff at the disposal of COVID-19 units. However, the definition of urgent procedure is a thorny task, made even more complicated, as the duration of the lock-down was unknown at that moment. In this regard, more than one protocol had been proposed by the orthopedic societies [5,8,21]. To select surgical procedures and admissions, we followed a protocol similar to the AAOS/DGOU guidelines for phase III [22,23]. Data presented in this study refer only to inpatients and the overall workload is underestimated, as outpatients’ duties have not been quantified. As already described by other papers, clinical follow-ups were guaranteed for outpatients with fractures during conservative treatment. Postoperative controls and selected critical cases were performed at the discretion of the physician, resulting in 15% of the outpatient activity in comparison with 2019 [16,24]. Nevertheless, patients with less severe conditions started to decline spontaneously in their attendance not only for consultations but even for surgical procedures [25]; the number of traumas decreased because of the diminished traumatic activities during lock-down [17]. A trauma network with reference centers to address ultra-specific tasks (tumors, infections and polytrauma) allowed the centralization of the most resource-draining cases and relieved the COVID-hub-hospitals. A similar approach was also successfully adopted in other countries which allowed resources to be minimized and kept the standards of care, even if a dramatic reduction of activities was mandatory. Concerning the follow-up of outpatients, it was strongly centralized to the hospitals as every medical consultation on the territory was extremely difficult: medical offices were closed, general practitioners overwhelmed by COVID-19 cases and the population had no other choice than the hospitals. In this way, COVID-19 and non-COVID-19 patients converged on the same structures [26].

The second essential task was to avoid the in-hospital propagation of SARS-CoV-2. Dedicated trauma COVID-19 wards were established in 83% of hospitals, the majority hosted two patients in a room and all centers established dedicated operating rooms. COVID-19 patients and those with a positive swab test were allocated to COVID wards. However, only 48% of the patients were screened with a swab test: at the beginning, the availability of swab-tests was so limited, they were set aside for the most critical settings. In this regard, nasopharyngeal swab tests for SARS-CoV-2 have not been routinely used to make the diagnosis, nor to start therapy, but to confirm it. Moreover, at the beginning of the first wave, the reliability of swab tests had to be tested on clinically positive patients. From this perspective, there were more critical situations than traumatology [27,28,29]. In the present study, only 48% of the patients with the diagnosis of COVID-19 in traumatology also had a positive swab test. According to our data, there were 7% patients with COVID-19 in traumatology during the worst scenario, circa 50% of them with a positive test; 7% tested positive for SARS-CoV-2, accounting for circa 14% patients to manage in COVID wards. These data could help to plan the division of wards and staff appropriately. During the lock-down, 34% of the cases were proximal femur fractures, almost all involving geriatric patients, who need a multidisciplinary approach, more resources and longer hospitalization; furthermore, the load for rehabilitation and long-stay centers during the lock-down could slow down the turn-over in traumatology wards. Some 13% (53/424) of patients with proximal femur fracture had COVID-19 and the mortality at 1 month in this group reached 23%, in contrast to 5% of COVID-19-free. As expected, a higher mortality in COVID-19 patients has already been reported by other Authors [17], although this is one of the biggest cohorts described by now. Particularly, in a similar study the Spanish HIP-COVID Investigation Group [30] observed 30% mortality and up to 67% of in-patients treated non-operatively. It was also proposed by Catellani et al. [31], that surgery might stabilize COVID-19 patients with proximal femur fractures, however, we still do not have enough data to give evidence-based recommendations. Attention and suspicion for carriers must be maintained in COVID-19 “free” environments, a constant education of the staff is mandatory [7]. In an ideal situation, every patient should be considered contaminated, but this is resource-consuming (rooms, PPE, etc.) and was not feasible during the red phase in our experience, due to the bed shortage. Indeed, a reliable screening method for asymptomatic carriers was not available, the critical site was the COVID-“free” wards.

Orthopedic and Trauma Departments provided staff for COVID-19 tasks; as many other specialists were reconverted in a few days for duties for which they were not trained [17,25]. The medical staff of 3/11 Departments of Orthopedics and Traumatology involved in COVID units were redirected to COVID units without specific training, the others had various formulas [26]. This reorganization exposed healthcare workers (HCWs) to medico-legal problems. The pandemic has exacerbated the problem of protecting health professionals, more than ever, from a distorted use of liability actions. As declared by D’Aloja et al. [32], the assumption is that, even in times of uncertainty, HCWs always have an answer, using universal guidelines to protect citizens. During the COVID-19 pandemic, those guidelines that are the parameter for evaluating the conduct of operators, cannot yet be defined. Hence a penal shield for the emergency was proposed by the government but it has never been approved. On one hand, the limitation of the surgical and outpatient clinical activities pushed the HCW to hardly select the pathologies to defer [33,34,35,36]; when both conservative and surgical treatments were possible (both in traumatology and orthopedics), there was an obvious pressure to choose the conservative treatment. This could surely lead to litigation, however, a few months after the 1st wave, at the time of completing the present survey, it is still the time of physiotherapy and rehabilitation, whilst the time of controversy is yet to come. On the other hand, HCWs from every field were reallocated in COVID-19-units, accomplishing duties they were not trained for. To meet the needs of their insured clients, the providers of the official collective insurance policy of the Italian Society of Orthopedics and Traumatology (SIOT-SAFE) [37] promptly granted explicitly the serious fault occurred during COVID-19 related duties. This was surely reassuring for the HCW, a commendable effort from the involved Insurance Companies and most companies soon followed the same path. According to the Italian jurisprudence, serious fault occurs in the case of neglectfulness, rough errors, inexperience, imprudence or malice. Most legal litigation in Orthopedics arise usually for nonserious fault, those cases being granted by dedicated insurance policies. Regarding the nonserious fault for orthopedics working in non-orthopedics areas there is still a lack of clarity. During an emergency, learning on-the-job is a necessity, however not acceptable as a plan: being prepared and trained is mandatory and a current educational program for the whole staff would be advisable. In addition, this would be useful not only for COVID-19, but for any similar situation [38,39]. In short, in terms of hospital reorganization, to anticipate problems was the first lesson learnt. The second was flexibility: during the red phase, Orthopedics and Traumatology Departments, together with other specialties, provided human resources for COVID-19 units [7,17,25,26] adapted to the different availability of resources and to new protocols, but a dramatic reduction of activities is mandatory.

### 4.2. Effects on the Medical Staff

Due to the heterogeneous organisations of the caregivers during phase 1 (often shared by necessity in more wards, facing tasks with various exposure risk), for an easier tracking of the exposure and duties in the present survey, only the medical staff was analysed. Of the staff, 11% orthopedic surgeons developed COVID-19, without known long-term consequences, 6% resulted asymptomatic carriers for SARS-CoV-2 on swab test, with 23 workdays lost on average.

The Italian Institute for Statistics (ISTAT), on behalf of the Italian Ministry of Health, conducted a seroprevalence study for SARS-CoV-2 at the end of the first wave (May−July 2020). The highest average prevalence of the general population was found in Lombardy (7.5%), with peaks in the provinces of Bergamo (24%) and Cremona (19%); in comparison, at the same time the Italian average prevalence was 2.5%. The same study also considered the prevalence among the health care workers, with a national prevalence of 5.6%, rising up to 9.8% in the region with a general prevalence higher than the national average. Of those seropositive for SARS-CoV-2, 27% had been asymptomatic, a further 23% had a few nonspecific symptoms [40,41]. As during the first wave, only the critical and hospitalized patients were tested, this could explain the discrepancy between the positive swab test performed during the first and critical wave and the number of cases detected during the following months. The prevalence of COVID-19 among the orthopedic surgeons in the present study is slightly higher than the means of the same region for all HCWs. This could be biased by the selection of the most involved centers.

The present study was not intended to find correlation of exposure and contamination, neither would it be possible with present data, as only 4/12 centers performed on their employees a screening at that time. However, the risk of contamination among patients and caregivers must be considered. Even if each case is one too many, the prevalence of COVID-19 among medical staff in the present study seems similar to other reports [7,10,24,42]. However, the hospital is a contagion hotspot and controlling the spread is challenging. On one side, the centralization of medical activities in a hospital causes a major concentration of cases, on the other side, the overload and fatigue for the staff increase the risk of exposure [3,42]. Haffer et al. [10] reported 2% of medical staff were infected with SARS-CoV-2 during the first wave in Germany, while the prevalence of the virus among the population and in hospital seemed lower [43] during that period. As during phase 1, the virus was diffused both in the territory and hospitals, the quantification of in-hospital risk of contamination is a challenging task, considering that a screening method for carriers is not available and the contact tracking often unreliable.

A shortage of PPE was perceived by circa 30% of the Departments and similar results have been reported in a worldwide survey for orthopedic surgeons [44]. This was not just a matter of hospital stock, but a nation-wide challenge of supply and demand; the inland production was not enough and importation from other countries was rationed, for the same reason: inland factories were reconverted, remembering old war times [45]. Along with adequate production and stock, caregivers’ compliance with protocols for PPE are paramount to avoid squandering [46].

Moreover, the risks of contamination must be considered both in COVID-19 areas and in “COVID-19-free” environments. Particularly in the latter, the risk could be underestimated by caregivers and it is paramount to remember that a COVID-19 free area is not SARS-CoV-2 free [47,48,49]: in the present study we could estimate around 7% of carriers among the asymptomatic patients while those values are probably an underestimation as the sensitivity of the swab test was about 70% [50,51]). In addition to the risk of physical contagion, another very important aspect to consider is the emotional risk. Health professionals had high levels of burnout and psychological symptoms during the COVID-19 emergency. A recent article by Giusti et al. [52], showed moderate to severe levels of emotional exhaustion and reduced personal accomplishment in more than 60% of the sample, and moderate to severe levels of depersonalization in more than 25% of the sample. In our study, the main concern was about infecting their families and colleagues, indeed 15 orthopedics decided to autonomously isolate themselves from relatives.

Cooperation with other services (infectious diseases, anesthesiology) was perceived as adequate in most centers; the common challenge made cooperation with other departments even more important, providing precious lessons in empathy and teamwork, looking beyond the routine.

Loss of work power must be considered when organizing the shifts [7]. Particularly in orthopedics, where usually teams are built with (fewer) redundant specific skills, independent and isolated sub-teams with the broadest spectrum of covered pathologies might preserve efficiency in the case of the contamination of one member [44]. No one had an instruction from the hospital either to avoid familiar contacts during the shifts in COVID units or to quarantine before coming back to “COVID-19-free” environments; this could be a further method to limit contamination in and out of hospital, as earlier described by Liang et al. [7]. In comparison with other HCWs, orthopedic surgeons are involved in procedures with the formation of bioaerosols, especially in operating rooms (among others, suspensions of water vapor from bone drilling, bone saws, combustion products from electrocauterization, ventilation procedures related to anesthesiology), as such they are a group at risk for contamination with SARS-CoV-2, similar to dentists, maxillofacial, plastic and spine surgeons [9]. The role of bioaerosols in orthopedic operating rooms and SARS-CoV-2 has been analyzed by Sharma et al. [53], concluding that the bioaerosol potential to transmit infectious diseases like COVID-19 is questionable. The rate of contamination observed reflects not only the spread of the disease, but also the surveillance policy during the first wave, hence it is biased towards underestimation: at that time, only those with symptoms were tested. Less biased in these terms is the number of HCWs affected by COVID-19, however it is impossible to distinguish between work or not-work related contamination [54].

### 4.3. The Orthopedic Point of View

The communication with government authorities was perceived as neutral or adequate, whilst with medical authorities as barely adequate in most centers. In Italy there is a regional-based health system, coordinated and overviewed by the Ministry of Health. Especially at the beginning of the first COVID-wave, each region answered independently and with different protocols; indeed, the situation in every land was very different [55] with some hot areas and some apparently free zones. Soon after, a national central crisis-unit coordinated the regional health systems [55]. In our opinion, the coordination inertia between the central/Regional Government and its operative application in a single institution through health authorities, together with no experience with such causalities and insufficient competence with COVID-19, led to unsure indications by the central crisis units. On the other hand, collaboration between different specialties was perceived as appropriate. Being prepared (e.g., with operative protocols at every level) could relieve the problem and as known by experience, practice exercises and drilling the skills are necessary to maintain the know-how and update procedures [56,57].

Circa 90% of the departments will not achieve their objectives in 2020, in particular patient-care objectives will not be achieved with a large increase in waiting lists and the distribution of assets in the single institution for the immediate/medium future could change by the perception of priorities. This is true overall for elective procedures, which are the most common ones in the EU/USA [10,43]; particularly in Italy, joint replacement procedures are the most common diagnoses after deliveries and procedures for myocardial pathologies [58]. Nevertheless, concerns about the financial consequences were described by a few centers, but it will apply only if the financial resources match the demand in public hospitals. At the same time, an overall drop in orthopedic and trauma activities and role perception is expected by most participants.

Concern about the perception of the role of Orthopedics and Traumatology arises in the following phases. This is with reference both to budget and perception of the hospitals’ objectives. Up to now, the system has not recovered, there are limitations in operating rooms, waiting lists for patients, hospital staff and services (anesthesiology, ICU, etc.) even for non-urgent but debilitating orthopedic pathologies [20,59].

The main limitations of this work are the limited number of participants during an emergency situation. This is an inter-regional survey, not national, but due to the spotty distribution of the pandemic at that time, grouping the prevalence data for the entire nation would have strongly biased the results. This is a picture of the “real life” scenario occurring during the red phase of the COVID-19 outbreak in one of the first and most affected areas around the world.

## 5. Conclusions

Orthopedics and trauma departments in Northern Italy were strongly affected by the COVID-19 pandemic, the work power was reduced by contamination or involvement in COVID units and only urgent procedures were performed; a reduction in non-urgent activities was necessary during the acute phase. The restructuring of hospital routine made cooperation with other departments even more important and provided precious lessons in empathy and teamwork, looking beyond the routine. The risk of contamination among patients and caregivers must be considered, overall, until a reliable screening method will be available; this risk was quantitatively predominant, in comparison to the management of COVID-19 patients with trauma. Preparation for such causalities is mandatory and the experience gathered in highly involved environments provides the evidence to build a coordinated global response and avoid the propagation of errors.

## Figures and Tables

**Figure 2 ijerph-18-05340-f002:**
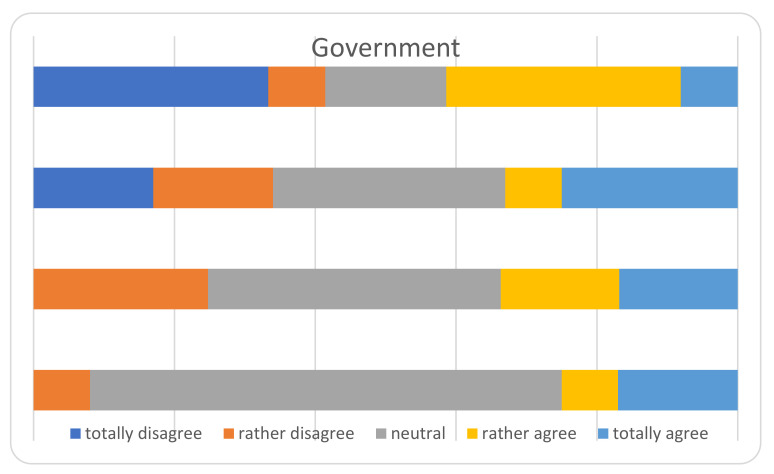
Perceived role of the government.

**Figure 3 ijerph-18-05340-f003:**
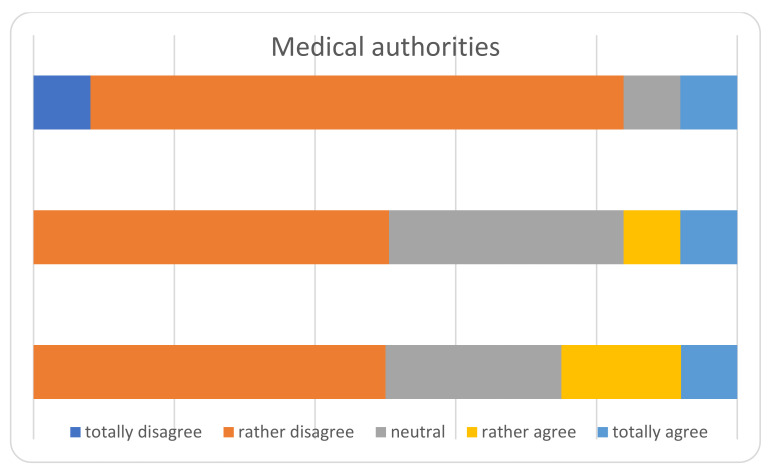
Perceived role of the medical authorities.

**Figure 4 ijerph-18-05340-f004:**
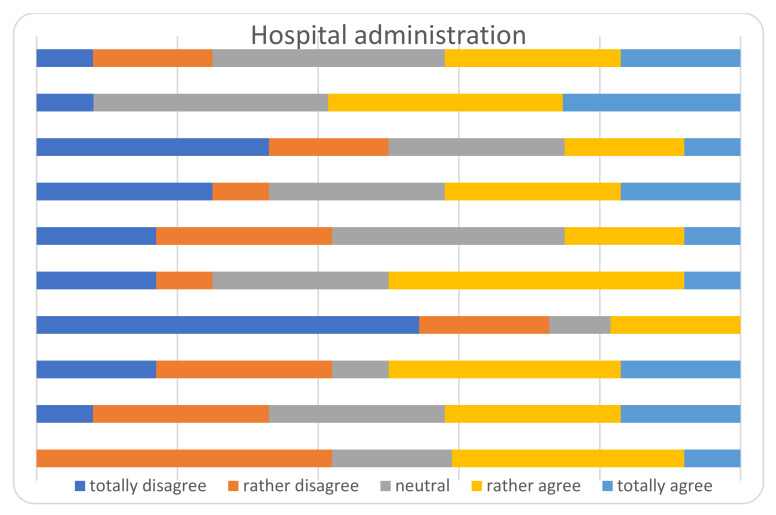
Perceived role of the hospital administration.

**Figure 5 ijerph-18-05340-f005:**
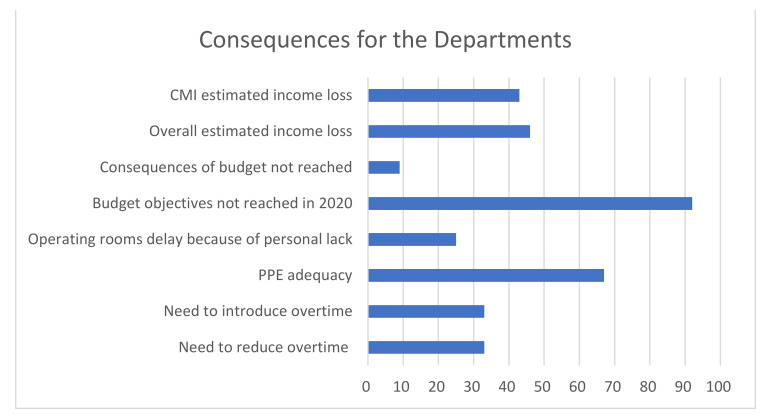
Consequences for the departments. CMI, Case Mix Index; it refers to the estimated income loss, according to the CMI. Each case is weighted, depending on its complexity, according to the diagnosis related group (DRG) [12,13]. PPE, Personal Protective Equipment.

**Figure 6 ijerph-18-05340-f006:**
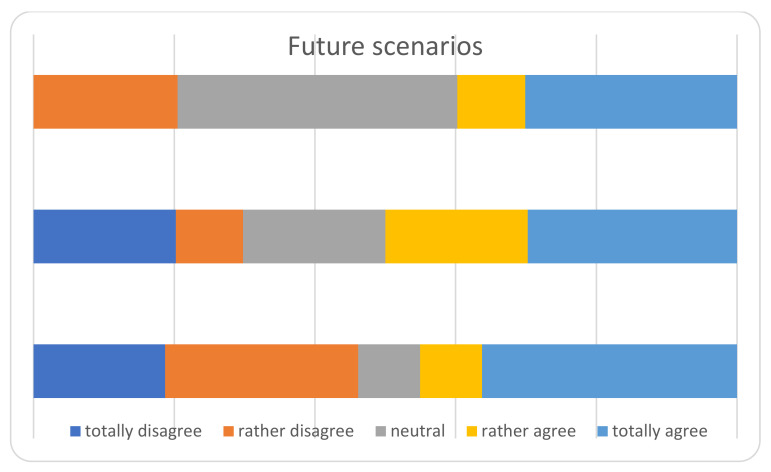
Future scenarios.

**Table 1 ijerph-18-05340-t001:** Characteristic of the centers involved in the present survey.

Centre	Type		Medical Staff (Nr)	Volume (No of Procedures) and Fraction with the Same Period in 2019	
				2020	2020/2019
1	3TC	AH	23	111	0.36
2	OI	UH/IR	96	302	0.10
3	3TC	UH	34	101	0.50
4	2TC	/	9	15	0.20
5	OI	UH/IR	10	95	0.10
7	OI	UH	6	50	0.70
8	3TC	UH	25	59	0.25
9	3TC	UH	17	62	0.05
10	3TC	UH/IR	18	137	0.50
11	2TC	/	13	37	0.30
12	3TC	UH	38	137	0.50
13	3TC	UH	21	67	0.30

Note: 14/14 centers collected data about patients (endpoint I: reorganization), 12/14 answered the survey (endpoints II and III: staff and subjective perception). 3TC, tertiary trauma center; 2TC, secondary trauma center; OI, Orthopedic Institute; AH, academic hospital; UH, university hospital; IR, biomedical institutions of relevant national interest and research activities (IRCCS).

**Table 2 ijerph-18-05340-t002:** Patients’ demographic data.

Variable	Raw	m or %	s	CI_95_
Total No.	1390			
M/F	510/1390	37%		34–39
Age		66	19	65–67
Femur (all)	713/1390	51%		49–54
Proximal femur	477/1390	34%		32–37
Age-proximal femur		81	0.6	81.38–81.48
No. of Pediatric ^a^ cases	82/1390	6%		5–7
NF-ST (all)	667/1390	48%		45–51
Positive NF-ST (all)	84/667	13%		10–15
NF-ST on admission	554/1390	40%		37–42
Positive NF-ST on admission	37/554	7%		5–9
COVID-19 patients	91/1390	7%		5–8
COVID-19 patients with positive NF-ST	44/91	48%		38–58
COVID-19 patients in ICU	13/91	14%		9–23

Note: m, mean of the sample; s, standard deviation of the sample; CI_95_, confidence interval (95%); NF-ST, nasopharyngeal swab test for SARS-CoV-2; ICU, intensive or sub-intensive care units, intended as a location for the invasive ventilation; ^a^ pediatric patients, age < 18 years.

**Table 3 ijerph-18-05340-t003:** Frequency of fractures by locations.

Fracture(s)	Location	Raw	Frequency	CI_95_
Humerus	All	190	0.14	0.12–0.16
	Proximal	68	0.05	0.04–0.06
Wrist		49	0.04	0.03–0.05
Femur	All	717	0.52	0.49–0.54
	Proximal	551	0.40	0.37–0.42
	Diaphiseal	49	0.04	0.03–0.05
Leg	All	202	0.15	0.13–0.16
	Diaphyseal	27	0.02	0.01–0.03
Pelvis	Acetabulum	36	0.03	0.02–0.04

## Data Availability

The data presented in this study are available on request from the correspondence author.

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
