# Peer review of "Clinical Activities, Contaminations of Surgeons and Cooperation with Health Authorities in 14 Orthopedic Departments in North Italy during the Most Acute Phase of Covid-19 Pandemic"

_ijerph, 2021, doi:10.3390/ijerph18105340_

Round 1

Reviewer 1 Report

I have read with great interest the work entitled  " Clinical activities, contaminations of surgeons and cooperation with health authorities in 14 orthopaedic departments in North Italy during the most acute phase of covid-19 pandemic". The paper is interesting but some changes are needed to make the paper publishable.

Major

  1. I find very interesting the times of "time to surgery" which despite the pandemic - for example for the femur - are very close to those indicated by international guidelines. For completeness, if available, I suggest a comparison with the pre-COVID timelines.
  2. With regard to the results, I believe it is important for the publication that the specific risks to which orthopedists have been subjected (for example following the execution of particular procedures) are highlighted compared to other specialties. [i.e. Nioi, Matteo, et al. "COVID-19 and Italian healthcare workers from the initial sacrifice to the mRNA vaccine: Pandemic chrono-history, epidemiological data, Ethical Dilemmas, and Future Challenges." Frontiers in Public Health 8 (2020).]
  3.  The problem of reorganization can often expose you to medico-legal problems. Have there been any complaints? Did orthopedic professionals work with peace of mind despite not being protected by a penal shield? [d'Aloja, Ernesto, et al. "COVID-19 and medical liability: Italy denies the shield to its heroes." EClinicalMedicine 25 (2020).]
  4. In order to have a better interpretation of the data, it seems useful to let the reader know what the personal protective equipment (PPE) were available for the HCWs, in your center.   in the different phases of the pandemic. This aspect should also be reported in the introduction. [Herron, J. B. T., et al. "Personal protective equipment and Covid 19-a risk to healthcare staff?." British Journal of Oral and Maxillofacial Surgery 58.5 (2020): 500-502].           
  1. Was the questionnaire used for the assessment validated?
  2. When we talk about "Effects on medical staff" it would be appropriate to introduce some statistics concerning the general population in the same period analyzed.

Minor

line 228: the reference in the square brackets is missing.

The use of footnotes at the end of the page is bizarre in a scientific article and - especially if you use numbers as references - it can be confusing. Authors should avoid the use of notes by summarizing what is reported in the text.

The small corrections suggested are important to make the paper publishable.

Reviewer 2 Report

This study presents a large cohort of orthopedics and traumatology patients and caregivers during the initial phase of the COVID-19 pandemic in northern Italy. I think that the work amassed relevant information, which could be useful to gain insight during this long ongoing crisis. I have a few observations.  

Minor Comments (by line number):

26 Correct the sentence: "From 10th March to 3rd May 2020 in Northern-Italy SARS-CoV-2 spread not contained, a disaster ‘triage was adopted."

40 The Abstract section must be rewritten, checking for style and grammar.

90 Please correct the sentence

163 Define OR

179 average

Please use point instead of comma as the decimal separator symbol (recurrent in many places of the text).

225 Main limitations of this work are :

313 The second

403 , and only urgent procedures

405 , and provided precious

408 would be available

410  evidence to build

Author Response

Dear IJERPH_”Reviewer 2”  

Thank you for having considered our study for IJERPH and your careful and precise review: we all Authors really appreciate it, along with the chance of improving our work.

Please see the reviewed manuscript in attachment.

Concerning your review: 

MINOR

26 Correct the sentence: "From 10th March to 3rd May 2020 in Northern-Italy SARS-CoV-2 spread not contained, a disaster‘triage was adopted."  Corrected

40 The Abstract section must be rewritten, checking for style and grammar. Corrected

90 Please correct the sentence Corrected

163 Define OR Corrected

179 average Corrected

Please use point instead of comma as the decimal separatorsymbol (recurrent in many places of the text). Corrected

225 Main limitations of this work are : Corrected

313 The second Corrected

403 , and only urgent procedures Corrected

405 , and provided precious Corrected

408 would be available Corrected

410 evidence to build Corrected

Best regards, 

The Authors

Round 2

Reviewer 1 Report

I congratulate the authors who have greatly improved the article following the directions given in the first round. Aside from asking for references 9 and 53 (duplicated) to be corrected, I have no further requests.

Author Response

Dear IJERPH_”Reviewer 1”

Thank you for having considered our study for IJERPH and your careful and precise review:  it and gave us a further chance of improving our work we all Authors really appreciate it.

The reference has been corrected.

Best greetings

Guindani Nicola
